# Ubiquitin Pathway Is Associated with Worsening Left Ventricle Function after Mitral Valve Repair: A Global Gene Expression Study

**DOI:** 10.3390/ijms21145073

**Published:** 2020-07-18

**Authors:** Feng-Chun Tsai, Gwo-Jyh Chang, Ying-Ju Lai, Shang-Hung Chang, Wei-Jan Chen, Yung-Hsin Yeh

**Affiliations:** 1Division of Cardiovascular and Thoracic Surgery, Chang-Gung Memorial Hospital, Taoyuan 333, Taiwan; lutony@cgmh.org.tw; 2College of Medicine, Chang-Gung University, Taoyuan 333, Taiwan; afen.chang@gmail.com (S.-H.C.); wjchen@cgmh.org.tw (W.-J.C.); 3Graduate Institute of Clinical Medical Sciences, Chang-Gung University, Taoyuan 333, Taiwan; gjchang@mail.cgu.edu.tw; 4Department of Respiratory Therapy, Chang-Gung University College of Medicine, Taoyuan 333, Taiwan; yingjulai@mail.cgu.edu.tw; 5Cardiovascular Department, Chang-Gung Memorial Hospital, Taoyuan 333, Taiwan

**Keywords:** ubiquitin, mitral valve, volume overload, heart failure

## Abstract

The molecular mechanism for worsening left ventricular (LV) function after mitral valve (MV) repair for chronic mitral regurgitation remains unknown. We wished to assess the LV transcriptome and identify determinants associated with worsening LV function post-MV repair. A total of 13 patients who underwent MV repair for chronic primary mitral regurgitation were divided into two groups, preserved LV function (N = 8) and worsening LV function (N = 5), for the study. Specimens of LV from the patients taken during surgery were used for the gene microarray study. Cardiomyocyte cell line HL-1 cells were transfected with gene-containing plasmids and further evaluated for mRNA and protein expression, apoptosis, and contractile protein degradation. Of 67,258 expressed sequence tags, microarrays identified 718 genes to be differentially expressed between preserved-LVF and worsening-LVF, including genes related to the protein ubiquitination pathway, bone morphogenetic protein (BMP) receptors, and regulation of eIF4 and p70S6K signaling. In addition, worsening-LVF was associated with altered expressions of genes pathologically relevant to heart failure, such asdownregulated apelin receptors and upregulated peroxisome proliferator-activated receptor gamma coactivator 1-alpha (PPARGC1A). HL-1 cardiomyocyte cells transfected with ubiquitination-related genes demonstrated activation of the protein ubiquitination pathwaywith an increase in the ubiquitin activating enzyme E1 (UAE-E1). It also led to increased apoptosis, downregulated and ubiquitinated X-linked inhibitor of apoptosis protein (XIAP), and reduced cell viability. Overexpression of ubiquitination-related genes also resulted in degradation and increased ubiquitination of α-smooth muscle actin (SMA). In conclusion, worsening-LVF presented differential gene expression profiles from preserved-LVF after MV repair. Upregulation of protein ubiquitination-related genes associated with worsening-LVF after MV repair may exert adverse effects on LV through increased apoptosis and contractile protein degradation.

## 1. Introduction

Severe, chronic, isolated mitral regurgitation (MR) is characterized by progressive dilation of the left ventricle (LV) due to significant volume overload.Severe MR also leads to a considerate remodeling of cardiac pathology including chamber dilation, cardiac fibrosis, and cardiomyocyte hypertrophy and apoptosis. The LV ejection fraction (EF) can be falsely normal due to the favorable condition for LV shortening dynamics when the LV myocardium has been irreversibly injured by severe MR. Surgery is the only promising therapy for severe MR and, as per the guidelines, an early surgery in symptomatic or asymptomatic patients is recommended before LV contractile function begins to deteriorate [1].

However, the timing of surgical correction of MR remains debatable [2,3]. There is a discrepancy in the data observed wherein early surgery is proved to be beneficial in some prospective and observational studies [3,4,5], whereas a watchful waiting strategy seemed to be safe and effective in another prospective study [6]. One concern with early mitral valve (MV) surgery is that it increases the likelihood of a second, redo MV surgery during the patient’s lifetime, since the valve may degenerate with time. In contrast, patients with severe chronic, isolated MR who received MV repair before LV EF declined, presented progressive LV dilation and worsening of LV EF after the surgery [4]. It also remains unclear when asymptomatic patients with severe MR with preserved LV EF should undergo surgical intervention [5]. Little was known about the mechanism of MR-related early myocardial damage before LV EF declined. In addition, there is currently no evidence-based pharmacological therapy aimed to prevent MR-related cardiac remodeling and heart failure. Unveiling the mechanisms underlying MR-causing irreversible LV function deterioration would improve the management of MR-related heart failure and help avoid early but unnecessary MR repair in asymptomatic patients.

Patients with normal LV EF before surgery may undergo LV remodeling and dysfunction after MV repair. The literature available describing LVremodelingafter repair of chronic MR is scarce and the mechanism is largely unclear [6]. It remains challenging to extend the survival of the patients with heart failure after surgical correction of the underlying cardiac disease. It may belie an incomplete understanding of heterogeneous mechanisms of heart failure due to diverse etiologies, including severe MR-related volume overload, and could be a barrier to a more precise treatment. A systemic approach using clinical specimens may help delineate specific genetic expression profile and pathway signatures contributing to pathologies.

Therefore, in this study, we explored the molecular mechanism of progressive LV remodeling despite mitral valve repair using a genome-wide approach. We revealed differential gene expression profiles in LV from patients with severe MR, with and without progressive LV remodeling after MV repair.

## 2. Results

### 2.1. Patient Characteristics

The clinical characteristics of patients are listed in Table 1. The age, pre-operative LV EF, and left ventricular end-diastolic diameter (LVEDD) were not significantly different between preserved-LVF and worsening-LVF. Echocardiography, six to nine months after MV repair, showed the post-operative LV EF values to be 64.9 ± 7.2% versus 48.6 ± 48.6% and LVEDD to be 45.5 ± 4.1 mm versus 52.2 ± 5.9 mm for preserved-LVF and worsening-LVF, respectively (Appendix A), and were significantly different between both groups. The data for the duration of severe MR before surgery between preserved-LVF and worsening-LVF were missing in some patients, hence, could not be compared.

### 2.2. Microarray

A total of 67,258 human expressed sequence tags (ESTs) were measured by microarray and 718 differentially expressed genes were identified (>25% fold-change) between worsening-LVF and preserved-LVF patients (*p* < 0.05), among which 138 genes were upregulated and 580 genes were downregulated (Appendix A). Figure 1A illustrates an unsupervised hierarchical clustering of changes in these genes between the two groups. Principal component analysis (PCA) verified the quality of microarray data. In this plot, samples of worsening-LVF and preserved-LVF were observed to be more similar within each group and apparently distinct between groups (Figure 1B). The volcano plot of the microarray is shown in Appendix A. Canonical cardiovascular pathway analysis showed activation of cardiomyocyte BMP receptor signaling and aldosterone signaling in worsening-LVF to be significantly enriched (Figure 2A). The right panel in Figure 2A and Appendix A demonstrate all the genes that were significantly upregulated in worsening-LVF. Figure 2B shows a set of pathways that were significantly activated or inactivated in worsening-LVF obtained after canonical intercellular and intracellular signaling analysis. Among them, 20 genes were involved in the protein ubiquitination pathway, which are differentially expressed between preserved-LVF and worsening-LVF, including 18 upregulated and two downregulated genes in worsening-LVF (Figure 2B, right panel and Appendix A).

### 2.3. Validation of Microarray with q-PCR

Microarray results were validated by q-PCR for selected genes shown in Figure 3 and genes related to the protein ubiquitination pathway. The results obtained showed a significant upregulation and downregulation consistent with the microarray data for the selected genes (Appendix A).

### 2.4. Functional Analysis of Microarray

Figure 3 illustrates a targeted association network of differentially expressed genes mapped in ingenuity pathway analysis (IPA), which were linked to cardiovascular diseases, including heart failure and cardiac arrhythmia. The expression of genes related to heart failure was altered in worsening-LVF. A significant upregulation of peroxisome proliferator-activated receptor gamma coactivator 1-alpha PPARGC1A, NPPB, and TNFAIP6 as well as downregulation of APLNR were observed, implying that pre-operative adverse LV remodeling was associated with worsening LV function after MV repair despite normal pre-operative LV EF. The expression of myocardial contractile elements such asMYBPC3 and MYH11 decreased in worsening-LVF, suggesting that contractile elements of LV myocytes may be compromised and underlie worsening LV function after MV repair. The circos plot of bio-functions and their corresponding genes significantly altered in worsening-LVF are shown in Appendix A.

### 2.5. Protein Ubiquitination in Cardiomyocytes

Since microarray identified genes related to the protein ubiquitination pathway (or ubiquitin-proteasome system, UPS) to be upregulated in worsening-LVF compared to preserved-LVF, we further evaluated the effects of these genes in cardiomyocytes. Plasmids containing cDNAs of the genes that were upregulated in worsening-LVF, namely USP15, PSMD7, UBE2D1, DNAJC15, DNAJC8, and pcDNA3.1, as their control were individually transfected into HL-1 cardiomyocytes. The efficacy of transfection was confirmed by Western blot (Appendix A).

Ubiquitin requires activation by UAE-E1 before ligation to substrate proteins, which are recognized and degraded by proteasomes. Therefore, we measured the expression level of UAE-E1 in the HL-1 cells using a Ubiquitin Human ELISA Kit 24-h after plasmid transfection. The expression of UAE-E1 was significantly upregulated in HL-1 cardiomyocytes transfected with plasmids containing USP15, PSMD7, UBE2D1, DNAJC15, and DNAJC8 compared with pcDNA3.1, suggesting that transfection with plasmid containing ubiquitination-related genes would significantly activate protein ubiquitination activity (Figure 4A).

### 2.6. Apoptosis and Intracellular Contractile Protein Degradation in Cardiomyocytes

Since worsening myocardial function may be attributed to loss of cardiomyocytes as well as to loss of intracellular contractile proteins, we further evaluated the effect of these ubiquitination-related genes on cardiomyocyte apoptosis and intracellular contractile protein degradation. The results obtained showed that both the apoptotic activity (Figure 4B) and myosin heavy chain (MHC) (Figure 6A) degradation were significantly increased, and cell viability (Figure 4C) was significantly reduced in cardiomyocytes transfected with USP15, UBE2D1, DNAJC15, and DNAJC8 compared with pcDNA3.1 control This suggests that UPS activation leads to increased apoptotic activity and a loss of contractile proteins in cardiomyocytes.

Caspase-3 and caspase-9 are proteases well known for their roles in carrying out apoptosis. X-linked inhibitor of apoptosis protein (XIAP) is one of the key proteins exerting anti-apoptotic activity by inhibiting caspase-3 and caspase-9. XIAP is known to be ubiquitinated and degraded by the UPS [7]. In HL-1 cardiomyocytes, the expression of active/cleaved caspase-3 relative to pro-caspase-3 was significantly increased in cells transfected with USP15, UBE2D1, and DNAJC15 compared with pcDNA3.1 control (Figure 4D).

The expression of active/cleaved caspase-9, relative to pro-caspase-9-, was significantly increased, and the expression of XIAP was significantly downregulated in cardiomyocytes transfected with USP15, UBE2D1, DNAJC15, and DNAJC8 compared with pcDNA3.1 control (Figure 4E and Figure 5A). These results suggest that the apoptotic activity was significantly increased by activated UPS.

To evaluate if XIAP and the contractile protein, α-smooth muscle actin (SMA), were labeled by ubiquitin, we performed co-immunoprecipitation to study the binding of XIAP and SMA with ubiquitin. The cell lysates were immunoprecipitated with anti-ubiquitin antibody and the expression of XIAP/SMA was evaluated by Western blot using an anti-XIAP and anti-SMA antibody. We performed Western blot for SMA and XIAP on the same membrane. The levels of total ubiquitinated proteins in HL-1 cardiomyocytes transfected with different plasmids were presented in Appendix A. The results obtained showed positive interactions between XIAP and ubiquitin, and SMA and ubiquitin in HL-1 cardiomyocytes transfected with USP15, UBE2D1, DNAJC15, and DNAJC8, suggesting activated UPS would cause degradation of XIAP and SMA in cardiomyocytes (Figure 5B and Figure 6B). Taken together, the findings suggest that the ubiquitination-related genes, which were upregulated in worsening-LVF, may activate the UPS and increase apoptosis as well as the degradation of contractile proteins in cardiomyocytes.

## 3. Discussion

### 3.1. Main Findings

In the present study, we identified differentially expressed genes in LV after MV repair in patients with worsening LV function as compared to patients with worsening LV function (worsening-LVF) with normal pre-operative LV function. The results of the microarray showed the expression of NPPB, which codes for brain natriuretic peptide (BNP), a well-known protein associated to decompensated heart failure, was around 15-fold higher in worsening-LVF compared with preserved-LVF (Figure 3). It implies that an adverse LV remodeling has already occurred in some patients with severe isolated, chronic MR, in whom the LV function continues to deteriorate even after MV repair. It is important to understand the mechanisms underlying adverse LV remodeling due to MR-related volume overload. In this study, we identified the activation of several genes and signaling pathways associated with worsening-LVF. The results obtained also suggest that activated UPS in worsening-LVF may contribute to cell apoptosis and loss of contractile proteins in cardiomyocytes, leading to progressive deterioration of LV function.

### 3.2. Chronic, Severe MR-Related LV Remodeling and Dysfunction

Chronic, severe MR causes LV volume overload, resulting in LV remodeling and leading to heart failure. Several signaling pathways and phenotypes have been reported to be altered in humans with chronic MR and animal models of volume overload, including activated β-adrenergic signaling, calcium-handling proteins [8], and downregulation of non-collagen extracellular matrix and profibrotic growth factor genes [9]. In a rat model of volume overload, the cardiac fibroblasts showed a hypoplastic phenotype [10]. Ahmet et al. reported that oxidative stress and disruption of cardiomyocyte desmin-mitochondrial sarcomeric architecture may be responsible for postoperative worsening LV function [11]. Recently, a study carried out in humans with chronic MR showed that transcriptional dynamics in LV endomyocardial biopsies correlated with adverse LV remodeling [12]. Chronic MR was associated with the altered expression of genes related to cell survival and extracellular matrix; decompensated chronic MR was associated with altered expression of SERCA2 and genes related to mitochondria, inflammation, extracellular matrix, and apoptosis. Following their findings, our study further identified several signaling pathways and molecules associated with progressive LV remodeling and heart failure in patients after MV repair, highlighting the role of protein ubiquitination-related genes.

### 3.3. Ubiquitin-ProteasomeSystem (UPS) and Heart Failure

The UPS is a major intracellular system responsible for the degradation of proteins, and plays a major role in regulating many cellular processes [13,14]. An increased abundance of ubiquitinated proteins and activated UPS is reported in humans and animal models of various heart diseases, including ischemic heart disease, dilated cardiomyopathy, and heart failure [14,15]. Our study presented activated UPS leading to loss of intracellular contractile proteins and apoptosis of cardiomyocytes, which could be associated with progressive LV remodeling and heart failure related to MR. Activation of UPS may contribute to pathological loss of intracellular proteins, therefore, the transcriptome profile may not necessarily corroborate with altered expression of proteins as well as their function in MR-related heart failure. In a mouse model of pressure overload, the proteasome inhibitor epoxomicin completely prevented cardiac hypertrophy while blocking proteasome activation [16], suggesting that the UPS is likely involved in the pathogenesis of heart failure.

It has been shown that the UPS regulates cardiac apoptosis [7]. Piacentino III et al. showed diminished expression of XIAP and increased apoptotic activity in humans with heart failure [17], which corroborates with our findings. It was shown that when the XIAP gene was expressed in rat neonatal cardiomyocytes, it attenuated apoptosis induced by protein kinase C inhibition, hypoxia/ischemia, or isoproterenol stimulation.

Ubiquitin-specific protease 15 (USP15), a member of cysteine protease deubiquitinases. It has been shown USP15 would remove ubiquitin from pro-caspase 3 [7], which may explain for increased expression of pro-caspase 3 and cleaved-caspase 3 (Figure 4D). The relationship between USP15 and ubiquitination of XIAP and α-SMA has not yet been clearly investigated. Our results intriguingly showed that USP15 overexpression increased ubiquitination of XIAP and α-SMA, which cannot be directly mediated by USP15. We speculate it could be indirectly mediated through other factors, for example, activated NF-κB or oxidative stress [18,19]. Further studies should be conducted for the underlying mechanism.

Taken together, our findings suggest both UPS and XIAP have the potential to be novel therapeutic targets in preventing the progression of MR-related heart failure.

### 3.4. Signaling Pathways and Molecules Associated with Worsening LV Function

We showed that the expression of apelin receptor (APLNR), a G-protein coupled receptor widely expressed throughout the heart, was downregulated in worsening-LVF. Targeting APLNR helps prevent heart failure resulting from pressure overload via suppression of angiotensin-converting enzyme expression and pathogenic angiotensin II signaling [20]. In humans, it was shown that apelin administration caused peripheral and coronary vasodilatation, and increased cardiac contractility in patients with heart failure [21]. Therefore, reduced APLNR may contribute to the progressive deterioration of LV function in worsening-LVF and may represent a potential therapeutic target. The expression of MYBPC3 and MYH11 was reduced in worsening-LVF, thus, implicating that loss of sarcomere, the basic unit of muscle contraction, may be the underlying pathological mechanism of worsening LV remodeling after MR repair. PPARGC1A, a transcription coactivator of nuclear receptors and metabolism regulator important in cardiac metabolism regulation [22], was upregulated in worsening-LVF. Decreased PPARGC1A activities or expressions contributed to pathological hypertrophy and reactivation of the fetal genetic program in chronicheart failure. Therefore, we speculate that the upregulation of PPARGC1A was compensatory in nature and exerted protective effects against MR-related adverse LV remodeling.

### 3.5. Limitations

The sample size of the study is small, and may not reflect the whole profile of genetic changes and the mechanism for worsening LV function after surgical mitral valve repair. The duration of follow-up echocardiography after MV surgery in the study is not long, which may not reflect the long-term alterations in the patients. Considering the heterogeneity of patients, future studies with bigger sample sizes will advance the findings. We did not perform quantitative protein analysis due to the unavailability of human specimens. The analysis of mRNA transcripts may not exactly reflect the expression of proteins and their function, as we showed, and altered UPS may enhance protein degradation.

## 4. Conclusions

We explored the different molecular signaling associated with worsening LV function in patients with previously preserved LV function after MV repair for the first time, whichmay help reveal novel potential therapeutic targets for heart failure.

## 5. Materials and Methods

### 5.1. Study Subjects

The study protocol was approved by the institutional review board of ChangGung Memorial Hospital. Patients with severe, chronic, isolated MR and LV EF >55% who underwent MV repair were included in the study. Between December 2013 and June 2015, 18 participants were recruited and LV specimens were obtained after patient consent. These patients were operated on by the same surgeon with a technical unit to avoid any deviation in the procedure. All 18 patients were followed up and the LV EF and diameter were evaluated by echocardiography up to six to nine months after surgical mitral repair. The patients following the criteria were recruited into two groups; group A included patients with improved or unchanged post-operative LV EF without increase in LV end-diastolic diameter (LVEDD) and group B included patients with reduced post-operative LV EF >10% with or without increase in LVEDD. The LV specimens of patients from both groups were collected for a microarray study. Patients who showed improved or unchanged LV EF but increased LVEDD were not included in the study. Therefore, of the 18 patients, eight and five patients were included in group A (preserved LV function, preserved-LVF) and group B (progressive LV remodeling, worsening-LVF) for furtherstudy, respectively.

### 5.2. Microarray Studies

The isolation of tissue RNA and its quality and quantity analysis were carried out as described previously [23]. Gene expression profiles in tissue RNA were analyzed using human HTA2.0 GeneChip (Affymetrix, Santa Clara, CA, USA), following the manufacturer’s protocol as described previously. The statistical and hierarchical clustering analyses and data visualization were performed as described previously [23].

### 5.3. Pathway and Network Analysis

Genes of interest were analyzed using ingenuity pathway analysis (IPA; Ingenuity Systems Inc, Redwood City, CA, USA). Fisher’s exact test was applied to examine the likelihood that the association between genes of interest and related pathways was not random. Functions that were predicted to be influenced by the differentially expressed genes were ranked according to their order of significance and were further analyzed by relevance to cardiovascular diseases or second messenger and intracellular signaling.

### 5.4. Quantitative Polymerase Chain Reaction (q-PCR)

TRIzol reagent was used for RNA extraction from the LV specimens, and q-PCR was performed as previously described [24]. Glyceraldehyde 3-phosphate dehydrogenase (GAPDH) mRNA was used as the internal control. The oligonucleotide sequences used are listed in Appendix A.

### 5.5. HL-1 Cell Culture

HL-1 cardiomyocytes were cultured in the Claycomb medium and subjected to field stimulation as described previously [25].

### 5.6. Expression Vectors and Transfection

A set of plasmids containing cDNAs of USP15, PSMD7, UBE2D1, DNAJC15, and DNAJC8P were purchased from OriGene (OriGene Technologies, Rockville, MD, USA). The vectors were transfected into HL-1 cardiomyocytes using Lipofectamine 2000 (Thermo Fisher Scientific Inc, Waltham, MA, USA) and used for experiments 24 h after transfection.

### 5.7. Determination of Ubiquitin Activating Enzyme E1(UAE-E1) Concentration

HL-1 cells were grown to confluence in 60 mm dishes. The concentration of UAE-E1 in the HL-1 cells was determined by a mouse E1/ubiquitin-activating enzyme ELISA kit (MyBioSource, San Diego, CA, USA) after plasmid transfection, as per the manufacturer’s instructions.

### 5.8. TUNEL Staining

Apoptosis was detected by TUNEL assay. After plasmid transfection, the HL-1 cells were grown in 24-well culture dishes for 24 h. The cells were then fixed on slides and TUNEL reaction mixture (Roche Applied Science, Mannheim, Germany) was added to the sections according to the manufacturer’s instructions, followed by incubation at 37 °C for 60 min. After removal of the TUNEL reagent, the slides were rinsed with PBS, and TUNEL-positive cells were evaluated using a confocal microscope (Leica TCS SP2, Wetzlar, Germany) with excitation at 488 nm with an argon laser. Emission was recorded using a Longpass> 550 nm filter set to acquire two-dimensional images (512 × 512 pixel).

### 5.9. Cell Viability Assay

The cell viability was measured by cell counting kit-8 (BioTools, Nangang, Taipei) according to the manufacturer’s protocol. Briefly, HL-1 cardiomyocytes were seeded into a 96-well plate at 37 °C with 5% CO2. Then, cells were transfected with different plasmids for 24 h. Afterwards, 10 μL of CCK-8 reagent was added to each well and incubated for 2 h at 37 °C. Finally, the absorbance at 450 nm was determined using a microplate reader (Bio-Rad Laboratories, Benicia, CA, USA).

### 5.10. Western Blot

Western blot was performed following the protocol as described previously [25]. Briefly, an equal amount of protein in sodium dodecyl sulfate-polyacrylamide gel electrophoresis (SDS-PAGE) sample buffer was sonicated and subjected to electrophoresis on 8% SDS-polyacrylamide gels. After being transferred to a PVDF membrane (Millipore, Temecula, CA, USA), proteins were incubated with primary antibodies against pro-caspase-3, cleaved caspase-3, pro-caspase-9, cleaved caspase-9, MHC, XIAP, SMA (Abcam, Cambridge, MA, USA), USP15, PSMA7, DNAJC8, DNAJC15, UBS2D1 (Abclonal, Woburn, MA, USA), and GAPDH (Santa Cruz, Dallas, TX, USA). Signals were detected by UVP (Analytik-Jena, Jena, Thuringia) and quantified by the software Image Gauge (Fuji film, Tokyo, Japan). GAPDH was used for the normalization of signal bands of other genes.

### 5.11. Co-Immunoprecipitation

The lysates from HL-1 cells were harvested with lysis buffer (25 mmol/L Tris-HCl-pH 7.6, 0.3 mol/L NaCl, 1.5 mmol/L MgCl_2_, 0.2 mmol/L EDTA, 0.5% Nonidet P-40, and 0.5 mmol/L dithiothreitol) and immunoprecipitated with anti-ubiquitinylated protein antibodies (Chemicon, Millipore, Temecula, CA, USA) for 2 h at 4°C. Following incubation with 50 μL of Protein A-Sepharose for 1 h at 4°C, beads were collected, washed three times, and resuspended in SDS sample buffer. The immunocomplexes were resolved by SDS-PAGE and analyzed by Western blot with anti-XIAP and anti-SMA (Abcam, Cambridge, MA, USA).

### 5.12. Immunohistochemistry

Immunohistochemical analyses were performed by confocal microscopy using myosin heavy chain (Abcam, Cambridge, MA, USA) primary antibody followed by Cy3 (Red, Chemicon, Temecula, CA, USA)-conjugated secondary antibody. Nuclei were counterstained using DAPI. Myosin degradation was quantified as the cytoplasmic myosin (MHC) area divided by the nuclear area. The relative expression levels of MHC were normalized to the control level. For each analysis, at least five fields were selected at random to observe >30 myocytes.

### 5.13. Statistical Analysis

For patient characteristics, mean ± standard deviation for continuous variables, and count and percent for categorical variables are presented, respectively. Fisher’s exact test was used for category comparisons between groups. For microarray analysis, the data were normalized to the expression of housekeeping genes like GAPDH for each set and the difference between the two groups was considered significant for a *p* value of <0.05 and relative fold-change >25%. The data in the figures are presented with box-and-whisker plots. Unpaired Student’s t test and one-way analysis of variance (ANOVA) with post hoc Tukey tests were applied for the two groups and multiple comparisons, respectively. A *p* value of <0.05 was considered statistically significant. The original data files of the microarray have been deposited in Gene Expression Omnibus (http://www.ncbi.nlm.gov/geo) with the access number GSE150661.

## Figures and Tables

**Figure 1 ijms-21-05073-f001:**
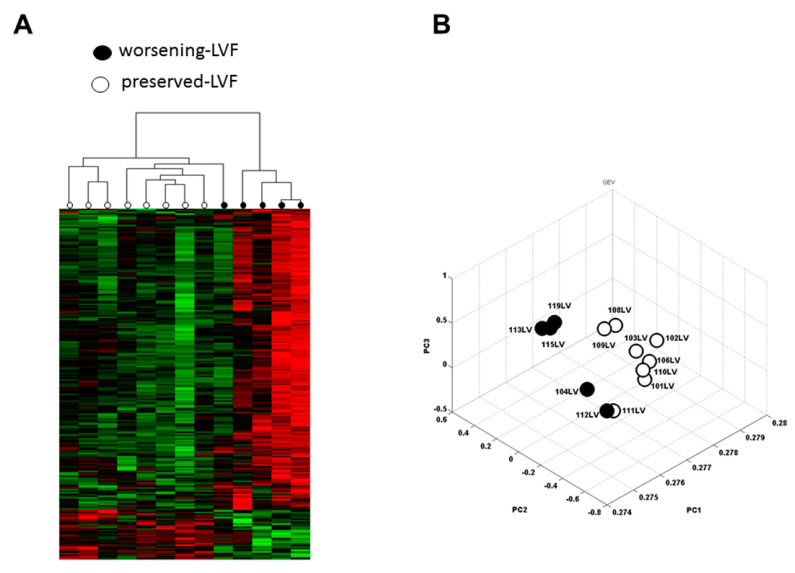
Heatmap and principal component analysis (PCA). (**A**) Unsupervised hierarchical clustering analysis of the gene expression profiles between worsening-LVF and preserved-LVF. Each row represents an individual sample, and each column represents a specific expressed sequence tag (EST). Bar color indicates gene mRNA expression level. The rows (13) of colored bars (718) represent the significant genes altered by more than 25%, between five worsening-LVF patients and eight preserved-LVF patients. The scale bar represents the intensity of expression (green: low expression, black: medium expression, and red: strong expression). The dendrogram on the top depicts the relationship between individuals. The dendrogram scale was calculated via the distance method (i.e., line length is inversely proportional to relatedness). (**B**) The PCA plot demonstrates the quality of the array. Each dot represents an expression profile of an individual sample plotted by the PCA score. ● = worsening-LVF; ◯ = preserved-LVF.

**Figure 2 ijms-21-05073-f002:**
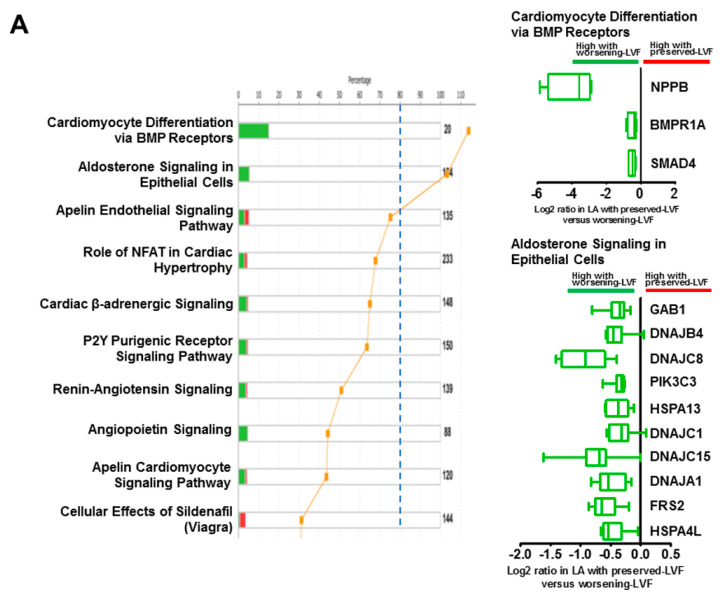
Altered signaling pathways identified byingenuity pathway analysis (IPA). (**A**) Stacked bar chart (left) shows activated canonical cardiovascular signaling pathways in worsening-LVF. Right panel represents altered genes involved in signaling pathways reaching threshold. (**B**) Stacked bar chart (left) shows activated canonical intercellular and intracellular signaling pathways in worsening-LVF. Right panel represents 20 genes involved in protein ubiquitination pathway in worsening-LVF. Data are represented as long 2 (worsening-LVF/preserved-LVF) in box-and-whisker plot.

**Figure 3 ijms-21-05073-f003:**
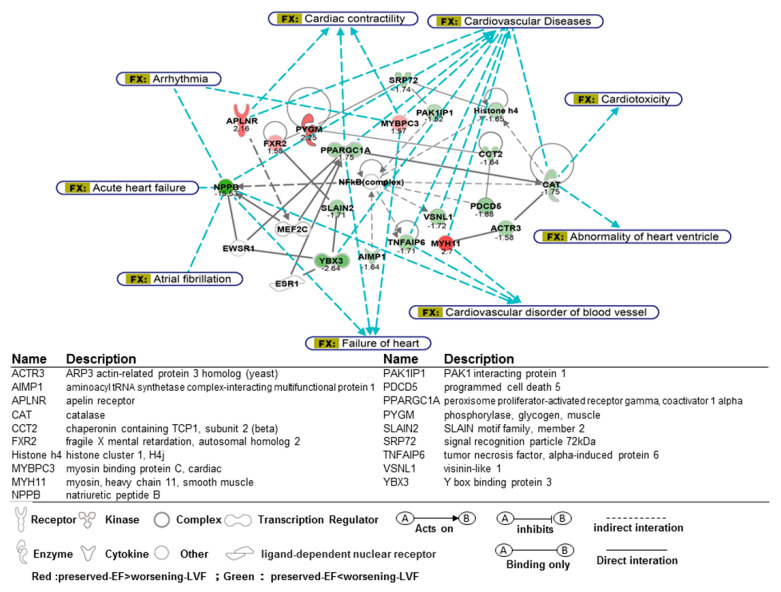
Network analysis for differentially expressed genes between worsening-LVF and preserved-LVF related to cardiovascular dysfunction and cardiac arrhythmia. A glossary for the gene symbols is shown in the lower panel. Red color indicates preserved-LVF > worsening-LVF. Green color indicates preserved LVF < worsening-LVF. Numbers represent fold-change.

**Figure 4 ijms-21-05073-f004:**
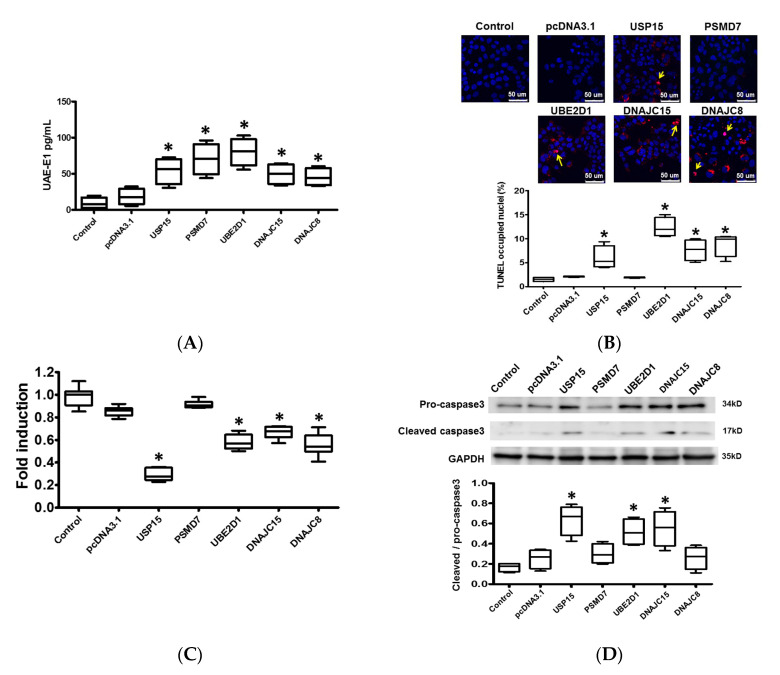
Effects of ubiquitination-related genes on protein ubiquitination and apoptosis in cardiomyocytes. (**A**) Box-and-whisker analysis of ubiquitin activating enzyme E1 (UAE-E1) expression in HL-1 cardiomyocytes transfected with USP15, PSMD7, UBE2D1, DNAJC15, and DNAJC8 and their pcDNA3.1 control by ELISA. * *p* < 0.05 versus control. N = 3 to 5 samples for each group. (**B**) Representative images of cell apoptosis by TUNEL assay. Increased green nuclear fluorescence reflects endo-nucleolytic DNA degradation and apoptosis of HL-1 cardiomyocytes. (**C**) Box-and-whisker analysis of cell viability by Cell Counting Kit 8 assay. The relative fold-change was normalized to the group of control. * *p* < 0.05 versus control. N = 3 to 5 samples for each group.(**D**) Representative blots and box-and-whisker Western blot analysis of pro-caspase 3 and cleaved caspase 3. (**E**) Representative blots and box-and-whisker Western blot analysis of pro-caspase 9 and cleaved caspase 9. The relative expression levels of proteins corresponding to GAPDH were quantified by densitometry and normalized to control. * *p* < 0.05 versus control. N = 3 to 5 samples for each gene.

**Figure 5 ijms-21-05073-f005:**
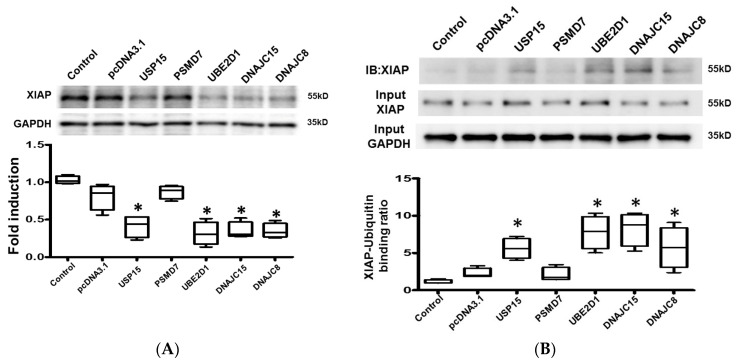
Effects of ubiquitination-related genes on X-linked inhibitor of apoptosis protein (XIAP). (**A**) Representative blots and box-and-whisker Western blot analysis of XIAP. (**B**) Representative blots and box-and-whisker Western blot analysis of XIAP immunoprecipitated with ubiquitin. The relative expression levels of proteins corresponding to GAPDH were quantified by densitometry. * *p* < 0.05 versus control. *n* = 3 to 5 samples for each experiment.

**Figure 6 ijms-21-05073-f006:**
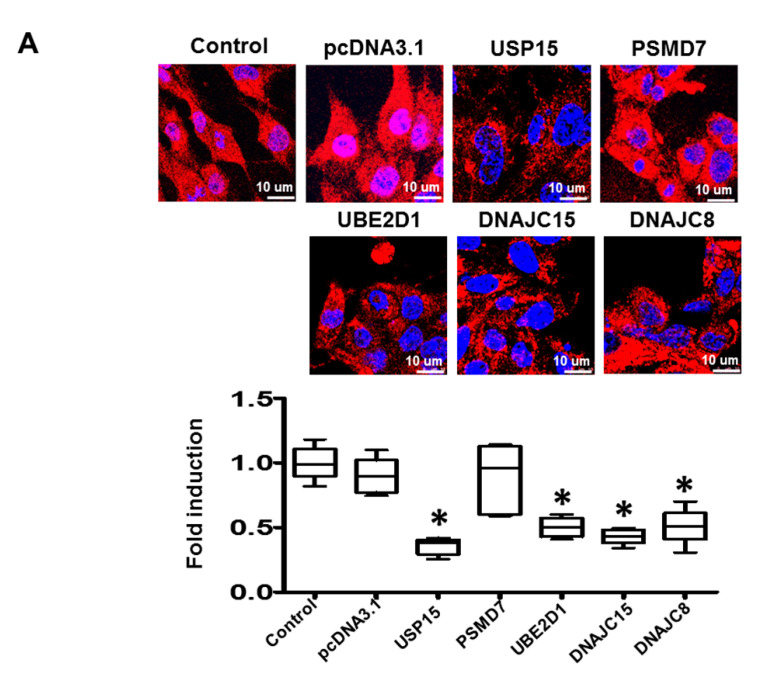
Effects of ubiquitination-related genes on degradation and ubiquitination of cardiac contractile proteins. (**A**) Representative images (upper panel) and box-and-whisker analysis (lower panel) of immunofluorescent staining of myosin heavy chain (MHC). Absence of red cytoplasmic fluorescence reflects MHC degradation in HL-1 cardiomyocytes. Red: MHC; blue: nuclei. (**B**) Representative blots and box-and-whisker Western blot analysis of α-smooth muscle actin (SMA) immunoprecipitated with ubiquitin. The relative expression levels of proteins corresponding to GAPDH were quantified by densitometry. * *p* < 0.05 versus control, *n* = 3 to 5 samples for each experiment.

**Table 1 ijms-21-05073-t001:** Demographic and clinical features of patients after maze procedure.

	Worsening-LVF (*n* = 5)	Preserved-LVF (*n* = 8)
Demographic Features		
Age (years old)	71.8 ± 14.0	62.4 ± 4.4
Sex, M:F	3:2	5:3
Clinical Features		
Pre-op LV EF (%)	63.6 ± 11.1	62.9 ± 9.9
Post-op LV EF (%)	50.4 ± 4.5	64.9 ± 7.2
Left atrialLA size (mm)	60.0 ± 9.6	49.0 ± 8.9
Rheumatic heart disease	1	4
Hypertension	0	2
Diabetes	0	0
Coronary artery disease	0	0

LVF, left ventricular function; EF, ejection fraction.

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
