# Peer review of "Ubiquitin Pathway Is Associated with Worsening Left Ventricle Function after Mitral Valve Repair: A Global Gene Expression Study"

_ijms, 2020, doi:10.3390/ijms21145073_

Round 1

Reviewer 1 Report

The authors has studies about the molecular mechanism for worsening left ventricular (LV) function after mitral valve (MV) repair for chronic mitral regurgitation. Of 67,258 expressed sequence tags, microarrays identified 718 genes to be differentially expressed between preserved-LVF and worsening LVF, including genes related to the protein ubiquitination pathway. Moreover they have demonstrated that HL-1 cardiomyocyte cells transfected with  ubiquitination-related genes activate the protein ubiquitination pathway with an increase in ubiquitin activating enzyme E1 (UAE-E1). Although this study is an interesting topic in the field, I have major concerns regarding the following issues.

-In Figure 5B and 6B, Gapdh lanes have extra lanes compared to other protein (IB: SMA and XIAP). Also, Gapdh protein expression patterns are quite similar between Figure 5B and 6B, are they identical? Did they perform IB for SMA and XIAP on the same membrane otherwise loading control should be prepared for each experiment.

-Additionally, the author should clarify why USP15 overexpression induce ubiquitination of SMA and XIAP proteins whereas USP15 is known as deubiquitinating enzyme (Figure 5B and 6B).

-Moreover, siRNA experiment is highly recommended to confirm the overexpression phenotype presented in Figure 4-6. The authors are dependent on the overexpression and it is better to confirm some result using siRNA approaches. Also, please check the supplemental data Figure S5 if Gapdh expression WB picture for internal control is not duplicated/reused in each experiments as they sometimes look quite similar.

Author Response

Reviewer 1.

The authors has studies about the molecular mechanism for worsening left ventricular (LV) function after mitral valve (MV) repair for chronic mitral regurgitation. Of 67,258 expressed sequence tags, microarrays identified 718 genes to be differentially expressed between preserved-LVF and worsening LVF, including genes related to the protein ubiquitination pathway. Moreover they have demonstrated that HL-1 cardiomyocyte cells transfected with  ubiquitination-related genes activate the protein ubiquitination pathway with an increase in ubiquitin activating enzyme E1 (UAE-E1). Although this study is an interesting topic in the field, I have major concerns regarding the following issues.

-In Figure 5B and 6B, Gapdh lanes have extra lanes compared to other protein (IB: SMA and XIAP). Also, Gapdh protein expression patterns are quite similar between Figure 5B and 6B, are they identical? Did they perform IB for SMA and XIAP on the same membrane otherwise loading control should be prepared for each experiment.

REPLY: Thanks for the reviewer’s comment. We changed Figure 5B and Figure 6B. We performed Western blot for SMA and XIAP on the same membrane and describe it in the manuscript. (Page 15 Line 225)

-Additionally, the author should clarify why USP15 overexpression induce ubiquitination of SMA and XIAP proteins whereas USP15 is known as deubiquitinating enzyme (Figure 5B and 6B).

REPLY: We thanks the reviewer pointed that USP15 is a deubiquitinating enzyme. We agree with the reviewer’s point and discuss it in the section of discussion as below. (Page 18 Line289)

Ubiquitin-specific protease 15 (USP15), a member of cysteine protease deubiquitinases. It has been shown USP15 would remove ubiquitin from pro-caspase 3,[7] which may explain for increased expression of pro-caspase 3 and cleaved-caspase 3 (Figure 4D). The relationship between USP15 and ubiquitination of XIAP and α-SMA has not yet been clearly investigated. Our results intriguingly showed that USP15 overexpression increased ubiquitination of XIAP and α-SMA, which cannot be directly mediated by USP15. We speculate it could be indirectly mediated through other factors, for example, like activated NF-κB or oxidative stress.[18, 19] Further studies should be conducted for the underlying mechanism.

Reference:

[7] I. Gupta, K. Singh, N.K. Varshney, S. Khan, Delineating Crosstalk Mechanisms of the Ubiquitin Proteasome System That Regulate Apoptosis, Front Cell Dev Biol 6 (2018) 11.

[18] Q. Zhou, C. Cheng, Y. Wei, J. Yang, W. Zhou, Q. Song, M. Ke, W. Yan, L. Zheng, Y. Zhang, K. Huang, USP15 potentiates NF-kappaB activation by differentially stabilizing TAB2 and TAB3, FEBS J  (2020).

[19] J. Qiu, T. Zhang, X. Zhu, C. Yang, Y. Wang, N. Zhou, B. Ju, T. Zhou, G. Deng, C. Qiu, Hyperoside Induces Breast Cancer Cells Apoptosis via ROS-Mediated NF-kappaB Signaling Pathway, Int J Mol Sci 21(1) (2019).

-Moreover, siRNA experiment is highly recommended to confirm the overexpression phenotype presented in Figure 4-6. The authors are dependent on the overexpression and it is better to confirm some result using siRNA approaches.

REPLY: We thanks the reviewer’s suggestion that experiment applying siRNA for ubiquitination pathway would further clear the role of ubiquitination in cell apoptosis and intracellular contractile protein degradation in cardiomyocytes. We think that although it was showed activation of ubiquitination pathway results to cell apoptosis and degradation of intracellular contractile proteins, it does not certainly indicate that inhibition of baseline protein ubiquitination would be protective against cell apoptosis and degradation of intracellular proteins in conditions without pathologic remodeling like volume overload. We agree it would be indicated to evaluate the effect of inhibition of protein ubiquitination pathway by siRNA in further studies, which, however, may not help explain a lot in the causal relationship between activated protein ubiquitination pathway and volume overload-related pathological remodeling.

Also, please check the supplemental data Figure S5 if Gapdh expression WB picture for internal control is not duplicated/reused in each experiments as they sometimes look quite similar.

REPLY: Thanks for the reviewer’s comment. We changes Supplemental Figure 5.

Reviewer 2.

The authors in the original research manuscript entitled “Ubiquitin pathway is associated with worsening left ventricle function after mitral valve repair: a global gene expression study” identified the molecular signatures associated with worsening LV function in patients who underwent MV repair for chronic primary mitral regurgitation. They also performed in vitro studies using HL-1 cells, to investigate the significance of observed regulatory molecules. They concluded that elevated ubiquitination-related proteins are associated with worsening-LVF after MV repair via increased apoptosis and degradation of the contractile protein. Overall, the study is well designed, easy to follow, and presented data are compelling. I have the following minor comments.

  1. Fig. 4 B, the authors used TUNEL assay (which measures DNA fragmentation in the nucleus) to detect apoptotic cells after transfection with various plasmids. I did not see any red fluorescence in nuclei (blue) in given representative images. It would be better if the authors determine the co-localization of red and blue colors using image analysis software and provide quantitative data.

REPLY: Thanks for reviewer’s comment. We changed Figure 4B, showing cells containing red fluorescence in nuclei and determine the co-localization of red and blue colors.

  1. The protocol for the cell viability experiment does not seem right. They transfected cells with plasmids in a 6-well plate, and cells were split and seeded in a 96-well plate 24 h after transfection. Cell viability was determined after 2 h. Observed differences might be due to improper trypsinization or cell loss during washing. It is advisable to investigate cell viability at least 24 h after seeding cells in a 96-well plate and it is not clear - how many cells were seeded/well of a 96-well plate?

REPLY: Thanks for reviewer’s comment. We revised the protocol for the cell viability. We revised it in sections of method and materials as below: (Page 24 Line 396)

The cell viability was measured by cell counting kit-8 (BioTools, Nangang, Taipei) according to the manufacturer’s protocol. Briefly, HL-1 cardiomyocytes were seeded into 96-well plate at 37°C with 5% CO2. Then, cells were transfected with different plasmids for 24 h. Afterwards, 10 μL of CCK-8 reagent was added to each well and incubated for 2 h at 37°C. Finally, the absorbance at 450 nm was determined using a microplate reader (Bio-Rad Laboratories, Benicia, California, USA).

  1. It would have strengthened the study if the authors have determined the levels of total ubiquitinated proteins in cells transfected with different plasmids. They can perform Western blot experiments using cell lysates and ubiquitin antibody.

REPLY: Thanks for the revewer’s suggestion. We add Supplemental Figure 6 presenting the level of total ubiquitinated proteins. (Page 15 line 226)

Reviewer 2 Report

The authors in the original research manuscript entitled “Ubiquitin pathway is associated with worsening left ventricle function after mitral valve repair: a global gene expression study” identified the molecular signatures associated with worsening LV function in patients who underwent MV repair for chronic primary mitral regurgitation. They also performed in vitro studies using HL-1 cells, to investigate the significance of observed regulatory molecules. They concluded that elevated ubiquitination-related proteins are associated with worsening-LVF after MV repair via increased apoptosis and degradation of the contractile protein. Overall, the study is well designed, easy to follow, and presented data are compelling. I have the following minor comments.

  1. Fig. 4 B, the authors used TUNEL assay (which measures DNA fragmentation in the nucleus) to detect apoptotic cells after transfection with various plasmids. I did not see any red fluorescence in nuclei (blue) in given representative images. It would be better if the authors determine the co-localization of red and blue colors using image analysis software and provide quantitative data.
  2. The protocol for the cell viability experiment does not seem right. They transfected cells with plasmids in a 6-well plate, and cells were split and seeded in a 96-well plate 24 h after transfection. Cell viability was determined after 2 h. Observed differences might be due to improper trypsinization or cell loss during washing. It is advisable to investigate cell viability at least 24 h after seeding cells in a 96-well plate and it is not clear - how many cells were seeded/well of a 96-well plate?
  3. It would have strengthened the study if the authors have determined the levels of total ubiquitinated proteins in cells transfected with different plasmids. They can perform Western blot experiments using cell lysates and ubiquitin antibody.

Author Response

(The authors gave the same response as above.)

Round 2

Reviewer 1 Report

N/A